# Facilitators and Barriers to Physical Activity and Sport Participation Experienced by Aboriginal and Torres Strait Islander Adults: A Mixed Method Review

**DOI:** 10.3390/ijerph18189893

**Published:** 2021-09-20

**Authors:** Bridget Allen, Karla Canuto, John Robert Evans, Ebony Lewis, Josephine Gwynn, Kylie Radford, Kim Delbaere, Justin Richards, Nigel Lovell, Michelle Dickson, Rona Macniven

**Affiliations:** 1Neuroscience Research Australia, Randwick, NSW 2031, Australia; ebony.lewis@unsw.edu.au (E.L.); k.radford@neura.edu.au (K.R.); k.delbaere@neura.edu.au (K.D.); 2Wardliparingga Aboriginal Health Equity, South Australian Health and Medical Research Institute, Adelaide, SA 5001, Australia; karla.canuto@sahmri.com; 3Faculty of Health and Medical Sciences, Adelaide Medical School, The University of Adelaide, Adelaide, SA 5005, Australia; 4School of Public Health, University of Technology, Ultimo, NSW 2007, Australia; john.evans@uts.edu.au; 5School of Population Health, UNSW Medicine & Health, UNSW Sydney, Kensington, NSW 2052, Australia; 6UNSW Ageing Futures Institute, University of New South Wales, Kensington, NSW 2052, Australia; n.lovell@unsw.edu.au; 7The Poche Centre for Indigenous Health, Faculty of Medicine and Health, The University of Sydney, Camperdown, NSW 2006, Australia; josephine.gwynn@sydney.edu.au; 8Faculty of Medicine and Health, The University of Sydney, Camperdown, NSW 2006, Australia; justin.richards@vuw.ac.nz (J.R.); michelle.dickson@sydney.edu.au (M.D.); 9School of Psychology, UNSW Science, Kensington, NSW 2052, Australia; 10Faculty of Health, Te Herenga Waka—Victoria University Wellington, Wellington 6012, New Zealand; 11Faculty of Engineering, Graduate School of Biomedical Engineering, UNSW Sydney, Kensington, NSW 2052, Australia

**Keywords:** indigenous, first nations, sport, exercise, population health, Australia

## Abstract

Physical activity has cultural significance and population health benefits. However, Aboriginal and Torres Strait Islander adults may experience challenges in participating in physical activity. This mixed methods systematic review aimed to synthetize existing evidence on facilitators and barriers for physical activity participation experienced by Aboriginal and Torres Strait Islander adults in Australia. The Joanna Briggs Institute methodology was used. A systematic search was undertaken of 11 databases and 14 grey literature websites during 2020. The included studies reported physical activity facilitators and barriers experienced by Aboriginal or Torres Strait Islander participants aged 18+ years, living in the community. Twenty-seven studies met the inclusion criteria. Sixty-two facilitators were identified: 23 individual, 18 interpersonal, 8 community/environmental and 13 policy/program facilitators. Additionally, 63 barriers were identified: 21 individual, 17 interpersonal, 15 community/environmental and 10 policy/program barriers. Prominent facilitators included support from family, friends, and program staff, and opportunities to connect with community or culture. Prominent barriers included a lack of transport, financial constraints, lack of time, and competing work, family or cultural commitments. Aboriginal and Torres Strait Islander adults experience multiple facilitators and barriers to physical activity participation. Strategies to increase participation should seek to enhance facilitators and address barriers, collaboratively with communities, with consideration to the local context.

## 1. Introduction

There are approximately 370 million First Nations people globally, with rich and unique cultures [1]. However, many First Nations people experience significant inequity in health outcomes [2]. Significantly divergent life expectancies between First Nations peoples and others in well-resourced countries, such as Canada, Australia, New Zealand and the United States of America, is testament to the ongoing impact of colonization [2]. These nations share common histories of marginalizing First Nations peoples through practices, such as genocide, dispossession and exclusion, whilst discrimination and racism remain ongoing structural determinants of health [3]. Alongside historical and structural determinants, social determinants, such as income, housing, employment, and education, also influence health outcomes and health behaviors, such as physical activity, for First Nations peoples [4]. Thus, any understanding of physical activity and health in First Nations populations must remain cognizant of these factors.

In Australia, Aboriginal and Torres Strait Islander people are the First Nations peoples, comprising many hundreds of culturally diverse nations [5]. Aboriginal and Torres Strait Islander people experience numerous health inequities, with a life expectancy gap of approximately 10 years less than non-Indigenous Australians [6]. There is a potential role for physical activity in reducing the burden of disease [7], and this has been identified as a priority issue by Aboriginal and Torres Strait Islander people [8]. A previous systematic review of physical activity interventions for Aboriginal and Torres Strait Islander people found evidence of improvements in participants’ metabolic profiles and quality of life, but no evidence of increased physical activity levels [9]. Physical activity and sport programs can also promote numerous social outcomes among Aboriginal and Torres Strait Islander people, including education, employment, social and emotional wellbeing, crime reduction and strengthened community and cultural ties [10].

There is no consensus on the definitions for “facilitator” or “barrier” in the context of physical activity, but both can be described as engagement factors related to behavior change, where facilitators are factors that enable, and barriers are factors that inhibit, participation [11,12]. Similarly, physical activity and sport can be difficult to define. Physical activity has been defined as “any bodily movement produced by skeletal muscles that results in energy expenditure” [13]. Sport and exercise can be considered subsets of physical activity, with exercise referring to structured, repetitive physical activity for the purpose of improving fitness [13] and sport referring to group or individual games with rules requiring motor action [14]. An international systematic review examining barriers and enablers to health behaviors in middle-aged adults found numerous factors influencing physical activity participation across personal, socio-cultural, physical, psychological and access factors [15]. Common barriers included lack of time, access issues, financial costs, entrenched attitudes, inappropriate environments, and low socioeconomic status [15]. Common facilitators included enjoyment of physical activity, motivation related to health benefits, social support and integration of physical activity into lifestyle [15]. A survey exploring the motivation of Australian adults found similar personal motivators, as well as desires to improve athletic performance and physical appearance, while the most commonly reported barrier was lack of time [16]. Facilitators and barriers can be studied by qualitative or quantitative methods. Recent qualitative research with Australian adults in rural settings identified environmental barriers, including lack of functionality for physical activity, pathway interruptions, and a lack of diversity of opportunities [17].

Physical activity plays a central role in Aboriginal and Torres Strait Islander cultures, including through traditional activities such as hunting and caring for Country; activities which not only require physical exertion, but which can hold great spiritual significance [18]. Early records suggest that at the time of European invasion, Aboriginal people were notably physically fit with lean body mass [19]. In contemporary Australia, Aboriginal and Torres Strait Islander children are more physically active than non-Indigenous children [6]. However, participation decreases through adolescence and only 38% of Aboriginal and Torres Strait Islander adults in non-remote areas are participating in sufficient amounts of physical activity [6].

A recent mixed-methods systematic review identified 37 facilitators and 58 barriers for Aboriginal and Torres Strait Islander children using a socio-ecological framework, across personal, interpersonal, community and policy domains [20]. Interpersonal factors played a significant role, with friends and family engaging in physical activity being an important determinant of participation, and the concept of “shame” being a particular deterrent. Children in remote areas often faced more barriers regarding access to facilities, activities, and transport and, in particular regions, uncomfortable weather patterns. A 2010 literature review, framed through the socio-ecological model, similarly identified the barriers of shame, lack of facilities and climate, as well as a lack of time and resources to support participation, that stemmed from socioeconomic disadvantage and family and work commitments [21]. However, physical activity was important in the lives of Aboriginal and Torres Strait Islander people, particularly for its role in social and community connections [21]. 

A systematic mixed methods review focusing on the adult population would give comprehensive understanding of facilitators and barriers to physical activity and sport for Aboriginal and Torres Strait Islander adults. Synthesis of this evidence base, using a socio-ecological framework, will provide insights into physical activity and sport participation and inform future program and policy design. This review aims to provide a synthesis of the current research exploring facilitators and barriers to participation in physical activity and sport experienced by Aboriginal and Torres Strait Islander adults.

## 2. Materials and Methods

This review uses the Joanna Briggs Institute mixed methods systematic review methodology [22] and is reported using the PRISMA statement for systematic reviews [23]. The full review protocol has been reported separately [24] and has been registered with PROSPERO (CRD42020214112).

The search was conducted between September and December 2020. A university librarian was consulted to advise on the search strategy. Keywords and index terms were developed for the four concepts of physical activity and sport, facilitators and barriers, Aboriginal and Torres Strait Islander people, and Australia. Eleven databases were searched: MEDLINE, CINAHL, EMBASE, Scopus, SPORTSDiscus, PsycINFO, Informit (ATSIhealth and AUSPORT), the Database of Abstracts and Reviews of Effects (DARE), the Cochrane Library, The Campbell Library and ProQuest Dissertations and Theses. Grey literature and websites were also searched, including Google Scholar, Australian Indigenous HealthInfoNet, Australian Institute of Aboriginal and Torres Strait Islander Studies, Australian Institute of Health and Welfare, Sport Australia, the Lowitja Institute and state and territory government sport and recreation documents. The search strategy was modified for these sites. For example, Google Scholar was searched with different combinations of “Aboriginal” or “Indigenous” and “sport” or “physical activity”. To limit results by relevance, the results were limited to “all in title”.

Studies were included if participants were Aboriginal or Torres Strait Islander adults living in Australia. Studies involving non-Indigenous adults or participants aged younger than 18 were considered if Aboriginal and Torres Strait Islander adult participants made up >50% of participants or if outcomes were reported separately. Studies describing community-based experiences were also included. Studies were included if they reported on facilitators, barriers, or related terms in relation to participation in all forms of physical activity and sport. Examples of “facilitator” synonyms include motivation or enabler, and synonyms for “barrier” include disincentive or obstacle. Facilitators or barriers needed to be reported in terms of the experiences of Aboriginal and Torres Strait Islander adults and not epidemiological correlates or observations by non-Indigenous participants. All structured and unstructured, group or individual programs were considered, and multi-component programs were considered for inclusion if physical activity and sport outcomes were reported separately. All states and regions of Australia were considered if participants were living in the community. All qualitative, quantitative, and mixed-methods studies were considered for inclusion. 

Studies that described non-contemporary experiences from before the year 2000 were excluded, as these were considered to be of less immediate relevance to current physical activity and participation. Experiences of elite athletes and sport coaches were excluded, as were programs delivered at live-in facilities and institutional settings, such as inpatient hospital programs. Reviews that examined physical activity and Aboriginal and Torres Strait Islander adults were checked for relevant studies. Only studies published in English were included. 

All study and website citations yielded were imported into Covidence [25] and duplicates were removed. All abstracts were screened using the inclusion and exclusion criteria by the first author, with the last author independently screening approximately 25% of the abstracts. The full texts of studies included after abstract screening were uploaded to Covidence and all were screened by the first author and independently crossed-checked by the last author. 

Data extraction was completed using a modified proforma from a previous Aboriginal and Torres Strait Islander physical activity and sport systematic scoping review [10]. For quantitative studies, phenomena data were in the form of data-based outcomes of descriptive or inferential statistical tests. For qualitative studies, phenomena data were in the form of themes or subthemes. 

Data were synthesized following a convergent integrated approach [26]. The extracted quantitative data were converted into sentence summaries and then assembled with the qualitative data based on similar meanings in order to form the identified facilitators and barriers. These findings were categorized using a modified version of the socio-ecological model of health [20]. Within each level of the model, findings were further subcategorized based on similarity, and subcategories were then used to form a set of integrated findings in the form of line of action statements.

### Critical Appraisal

The included studies were critically appraised by two independent reviewers for methodological quality using the Mixed Methods Appraisal Tool (MMAT) [27]. The included studies were also assessed for the quality of health research from a First Nations perspective prior to inclusion in the review using the Aboriginal and Torres Strait Islander Quality Appraisal Tool (QAT) [28]. Any disagreements that arose between the reviewers were resolved through discussion, or with a third reviewer where necessary.

## 3. Results

### 3.1. Study Inclusion

A total of 1423 studies were identified, and 1031 studies remained after duplicates were removed. After abstract screening, 102 full texts were retrieved for review. Ultimately, 27 studies were included (Figure 1). Three studies [29,30,31] were different analyses of the same data set and were grouped together for data extraction and presentation. Four studies were identified in grey literature [18,32,33,34].

### 3.2. Methodological Quality 

The methodological quality of studies was generally high when assessed with the MMAT [27], with 19 of the 27 studies scoring “Yes” for all five assessment categories (Appendix A). However, the quality of studies using the Aboriginal and Torres Strait Islander QAT was much poorer or unclear [28] (Appendix A). No studies scored “Yes” for all 14 categories. The highest scoring study scored “Yes” in 11 of the 14 categories [35], followed by a study scoring 10/14 [36] and five further studies scoring 7/14 [18,32,37,38,39]. All studies included scores of “Partial” and “Unclear” in several categories. No studies were excluded on their quality scores from either tool, as the appraisal results were considered to provide important lessons for the design and conduct of future studies. 

### 3.3. Characteristics of Included Studies

Table 1 details the characteristics of the included studies. The included studies were published between 2008 and 2020. The studies took place across all Australian states and territories, with six studies taking place across multiple states and territories [29,30,31,33,40,41]. Eight took place solely in Queensland (Qld) [32,36,42,43,44,45,46,47], two in Victoria [38,48], two in New South Wales (NSW) [49,50], three in the Northern Territory (NT) [18,35,51], three in Western Australia (WA) [34,37,52], one in South Australia (SA) [39], and one in Tasmania (Tas) [53]. One study did not report a location [54]. Nine studies took place in multiple geographic contexts [29,30,31,33,37,40,41,48,50]. Nine took place solely in urban settings [32,34,38,39,42,44,45,49,53], two in rural/regional settings [46,47] and seven in remote settings [18,35,36,43,51,52,54]. 

Four studies did not specify individual participant numbers [18,33,40,43]. Of the remaining studies, there were a total of 722 Aboriginal and Torres Strait Islander participants, and more male (56%) than female (44%) participants. Three studies included men only [41,52,54], while nine studies included women only. [18,29,30,31,32,39,40,43,50] Seven studies specified Aboriginal participants only [18,34,35,37,51,52,53], and two others included First Nations participants residing in the Torres Strait Islands or Northern Peninsula area of Queensland [36,43]. 

There were 21 qualitative studies [29,30,31,32,33,34,35,37,38,40,41,42,43,44,45,46,47,48,50,51,52,54] and six mixed-method studies [18,36,39,49,50,53], although only the qualitative data included facilitators and barriers in four of those studies. Most studies were cross-sectional, with only two longitudinal studies [32,50]. A variety of theoretical frameworks derived from health literature were utilized in many of the studies for analysis [29,31,32,35,37,38,39,41,43,44,46,47,48,52,54]. 

Nine studies involved structured physical activity and exercise programs [34,36,39,40,44,45,47,53,54], five had a sport focus [29,30,31,33,38], and ten examined broad physical activity perspectives [32,35,37,41,42,43,46,48,51,52]. Six studies were multi-component programs involving both physical activity together with other aspects, like nutrition [34,39,44,45,48,53]. One study specifically explored On-Country cultural activities, such as gathering traditional foods, as a form of physical activity [18]; one examined perspectives on exercise specifically [49]; and one study was a self-directed physical activity program involving structured or unstructured activity [50].

Six studies used participatory methods [18,34,35,41,50,51], of which three [34,41,50] used First Nations methodologies, such as Yarning—which is conversational storytelling [55]. Eight other studies used First Nations methodologies together with other ways of involving local communities [29,30,31,37,45,48,49,52]. One study used Yarning but did not otherwise specify local Aboriginal and Torres Strait Islander involvement [54]. A further eight studies involved local Aboriginal and Torres Strait Islander communities, but did not specify participatory or First Nations methodologies [32,33,36,38,39,43,53,54]. Overall, 21 studies involved local Aboriginal and Torres Strait Islander communities, of which nine collaborated with local community organizations or members [34,35,37,41,43,45,49,51,52]. There were eleven studies that used a consultation process with Aboriginal and Torres Strait Islander organizations or community members [29,30,31,34,35,37,41,49,51,52,54]. The remaining included studies did not specify local Aboriginal and Torres Strait Islander involvement. There were Aboriginal or Torres Strait Islander members of the research team as identified authors in 19 of the 27 studies [18,29,30,31,32,35,36,37,38,39,41,42,44,46,48,49,50,51,52].

### 3.4. Findings of the Review

Facilitators and barriers were roughly reported on in similar numbers, with findings of 23 individual, 18 interpersonal, 8 community and environment, and 13 policy and program facilitators (62 total), and 21 individual, 17 interpersonal, 15 community and environment and 10 policy and program barriers (63 total). However, when examined by geographic locations, more facilitators and barriers were identified in urban locations, most markedly at the policy and program level. Physical activity and sport facilitators and barriers are presented in Table 2, categorized by a modified version of the socio-ecological model of health used in the review of young people [20].

**Table 1 ijerph-18-09893-t001:** Summary characteristics of included studies reporting physical activity and sport facilitators and barriers experienced by Aboriginal and Torres Strait Islander adults.

Reference (Author, Date)	Aim	Design	Program/ Context	Data Collection Methods	Data Analysis	Theoretical Framework/Epistemology	Type of Physical Activity or Sport	State and Geographic Context	Participants		
									Number (*n*)	Sex	Age (years)
Andrews 2013 [34]	Evaluate an Aboriginal health & wellbeing intervention	Participatory action research, photovoice, semi-structured interviews, focus groups	Weekly free physical activities, healthy meal & “yarn” led by Aboriginal sports association	Interviews and focus groups, written notes	Thematic analysis	Not specified	Structured physical activity program: Zumba	WA Urban	*n* = 13	M: 3 F: 10	25–75
Canuto 2013 [39]	Identify perceived barriers & facilitators to attendance of Aboriginal & Torres Strait Islander women	Qualitative: individual semi-structured interviews	Cohort from pragmatic randomised control trial of a physical activity and nutritional education program	Interviews	Thematic network analysis	Chen’s program planning framework and socio-ecological frameworks (First Nations adapted)	Structured twice weekly exercise class	SA Urban	*n* = 16	F	18–64
Caperchione 2009 [40]	To report the barriers, challenges and enablers to physical activity participation in priority women’s groups	Focus groups	Community walking groups	Recorded and transcribed interviews	Inductive thematic analysis	Not specified	General, walking groups	VIC, NSW, QLD Urban Rural	3 Aboriginal and Torres Strait Islander focus groups	F	Not clear
Carr 2019 [35]	To explore perspectives of individuals and family on “what is important” and “what works best” to keep people with cerebellar ataxia walking and moving around	First Nations and participatory methodology, semi-structured interviews	N/A	Interviews and transcribed field notes	Inductive thematic analysis	Constructivist grounded theory	Walking and general	NT Remote	*n* = 12	M: 4 F: 8	30+
Cavanagh 2015 [54]	To identify impact of sport and active recreation programs on health, attitudes and behaviours, social connectedness, and sense of belongingness	Semi-structured individual interviews and Yarning circles	Pilot sport and recreation program combined with healthy eating program	Manually collected interview data	Thematic content analysis	Sarason’s framework of belongingness	Sports, recreational exercise	Not specified Remote	*n* = 9	M	Not clear
Davey 2014 [53]	To report on participation and effectiveness of a combined cardio-vascular and pulmonary rehabilitation and secondary prevention program	Qualitative: participant evaluation survey	8-week cardiopulmonary rehabilitation program, with bi-weekly exercise sessions and weekly educational sessions	Evaluation forms completed by participants at end of program	Iterative thematic analysis	Not specified	Rehabilitative exercise program	TAS Urban	*n* = 51	M, F	18+
David 2018 [18]	To find out the health benefit of self-initiated On-Country activities	Qualitative: participatory methods, oral conversations	N/A	Not specified	Not specified	Not specified	On-Country activities	NT Remote	Not specified	F	Not clear
Hunt 2008 [42]	To explore the meaning of, barriers to, and potential strategies to promote physical activity participation	Focus groups	N/A	Recorded and transcribed interviews	Iterative, thematic analysis	Not specified	General	QLD Urban	*n* = 96	M: 45 F: 51	18+
Lin 2012 [37]	To gain an in-depth understanding of chronic lower back pain experience	In-depth semi-structured informal interviews using Yarning	N/A	Recorded and transcribed interviews, additional data: field observation and notes from informal yarns	“Describe-compare-relate” process	Qualitative interpretive framework	General	WA Rural Remote	*n* = 32	M: 21 F: 11	26–72
Macdonald 2012 [43]	To examine discourse of lifestyle recruited to normalise living standards of Indigenous Australians, particularly women & girls	From larger study: semi-structured and open interviews with families, field notes and document collection	Cross-organizational network efforts to decrease diabetes through increasing recreational physical activity opportunities	Recorded and transcribed interviews	Iterative thematic analysis	Postcolonial critique	General	QLD Remote	21 families	F	Not clear
Macniven 2020 [49]	To identify exercise motivators, barriers, habits and environment	First Nations or “Indigenist” research methods, online survey	N/A	Online survey through survey monkey	Descriptive statistics with difference testing by Indigenous status	Not specified	Exercise	NSW Urban	*n* = 167 Aboriginal and Torres Strait Islander *n* = 45	M: 28 F: 17	18–64
Macniven 2018 [36]	To examine perceptions of the health and community impact of the Indigenous Marathon Program on Thursday Island	(a) Qualitative: semi-structured interviews (b) Quantitative: questionnaire	Indigenous Marathon Program: annually a squad of 12 young Indigenous adults aged 18–30 are selected to train to run a marathon while living in community	Recorded and transcribed interviews, paper questionnaire	(a) Thematic content analysis. (b) Descriptive statistics with difference testing by Indigenous status	Not specified	Running	QLD Remote	(a) *n* = 18 Aboriginal and Torres Strait Islander *n* = 14 (b) *n* = 104 Aboriginal and Torres Strait Islander *n* = 43	(b) M: 13 F: 29	18+
Maxwell 2019 [50]	To explore how digital health technologies contribution to Indigenous Australian women’s increased participation in physical activity	Qualitative: digital tracking and diarizing of activity levels for 8 weeks, focus groups/Yarning circles, individual interviews	Individualised self-designed health and activity goals, digitally tracked activity levels for 8 weeks	Recorded and transcribed interviews	Typological analysis	Not specified	Leisure activities	NSW Urban Rural	*n* = 8	F	18+
Mellor 2016 [41]	To record views on factors that contribute to poor physical health	Participatory action framework, focus groups, individual interviews	N/A	Recorded and transcribed interviews	Iterative analysis	Health belief model	General	VIC, WA Urban Rural Remote	*n* = 150	M	18–35
Nalatu 2012 [32]	To understand the physical activity needs and experiences of post-natal women	Follow-up case studies, in-depth interviews	N/A	Recorded and transcribed interviews	Thematic analysis	Theory of planned behaviour	General	QLD Urban	*n* = 27 total *n* = 10 Aboriginal and Torres Strait Islander	F	24–32
Nelson2016 [44]	To explore client and staff perception of the Work It Out program	Semi-structured small group or individual interviews	Chronic disease self-management and rehabilitation program: education session and tailored exercise session 2–4 times/week, optional individual meetings with allied health professionals	Transcribed interviews	Thematic analysis	Constructionism and Most Significant Change theory	Structured exercise program	QLD Urban	*n* = 22 Aboriginal and Torres Strait Islander *n* = 16	M, F	21–77
Parmenter 2020 [45]	To explore perceptions of the factors that influence their participation in rehabilitation program	Focus groups using Yarning, strengths-based approach	12-week cycle with twice weekly Yarning (education) and tailored supervised exercise	Recorded and transcribed focus groups	Inductive analysis	Not specified	Structured exercise program	QLD Urban	*n* = 102	M: 43 F: 59	18–80
Peloquin 2017 [46]	To determine regionally-based Indigenous Australian adults barriers or facilitators to PA, compared to regionally-based non-Indigenous Australians	Individual interviews	N/A	Recorded and transcribed interviews	Thematic analysis	Phenomenology	General	QLD Rural/ regional	*n* = 24 Aboriginal and Torres Strait Islander *n* = 12	M: 5 F: 7	18–55
Seear 2019 [52]	Identify how and why some Aboriginal people have made positive lifestyle changes	Individual interviews using Yarning	N/A	Recorded and transcribed interviews	Thematic analysis	Phenomonology	General	WA Remote	*n* = 4	M	20–35
(i) Stronach 2015 [31] (ii) Stronach 2018 [30] (iii) Stronach 2019 [29]	(i) To explore meaning, place and experience of sport & physical activity to Indigenous women, their needs and wants, and contribution to their health and well-being. ( ii) To explore sporting experiences and community strengths of Indigenous women (iii) To consider significance of swimming for Aboriginal women	Dadirri methodology, group or individual interviews using a “conversation approach”	N/A	Recorded and transcribed interviews	Inductive and deductive thematic analysis guided by Dadirri methodology	(i) Bourdieu’s social theory (ii) Empowerment (iii) Not specified	Sport and physical activity, swimming	NSW, TAS Urban Remote	*n* = 22	F	18–74
Sushames 2017 [47]	To explore barriers and enablers to participation in a community-tailored physical activity intervention	Semi-structured individual interviews	8-week free physical activity program	Recorded and transcribed interviews	Thematic analysis	Health belief model	Structured physical activity program	QLD Rural/ regional	*n* = 12	M, F	18–45
Thompson 2013 [51]	To explore local perspectives, experiences and meanings of physical activity	Participatory action research, semi-structured individual interviews,additional data sources: 5 local paintings and field observations	N/A	Recording and transcription of interviews, field observations recorded in journal	Thematic analysis of interviews and paintings	Not specified	General	NT Remote	*n* = 23	M: 9 F: 14	16–65
Thorpe 2014 [38]	To examine barriers and motivators for participation in Aboriginal community sporting team	Focus groups, semi-structured individual interviews	N/A	Recorded and transcribed interviews	Qualitative analysis	Grounded theory	Football	VIC Urban	*n* = 14	Not clear	Not clear
Walker 2020 [48]	To explore meaning of being healthy and how social media influences health behaviours	Face-to-face or online group semi-structured interviews informal Yarning with participant by phone or in person prior to interview	N/A	Recorded and transcribed interviews	Thematic analysis	Integrated model of behaviour change theory	General	VIC Urban Rural	*n* = 18	M: 9 F: 9	17–24
Young 2018 [33]	To research sports participation and physical activity behaviour and its context, patterns and drivers	Focus groups, in-depth individual interviews	N/A	Not specified	Not specified	Not specified	General	NT, SA, QLD, VIC, ACT, NSW Urban Rural Remote	38 focus groups 32 in-depth interviews	M, F	18+

New South Wales: NSW; Queensland: QLD; South Australia: SA; Tasmania: TAS; Victoria: VIC; Western; Australia: WA; Australian Capital Territory: ACT; Northern Territory: NT.

Table 3 presents the facilitators and barriers to physical activity and sport participation experienced by Aboriginal and Torres Strait Islander adults, organized by geographic location. These findings are summarized and described, and interwoven with the five Action Statements.

### 3.5. Individual Facilitators and Barriers

Action Statement 1: Personal attitudes and life circumstances of Aboriginal and Torres Strait Islander people should be considered in effectively identifying and addressing opportunities for physical activity and sport and potential barriers.

Sub-categories: Self-beliefs and attitudes, physical activity aligned with daily life and personal circumstances.

Numerous attitudes, expectations and self-beliefs were identified that could either facilitate or hinder participation. A multitude of personal circumstances influencing participation were identified, including health issues, socioeconomic factors, physical activity aligning with other life activities, and competing commitments. In urban studies, intrinsic self-motivation [32,39,49] facilitated participation, as did numerous personal motivators, including poor body image perception, [32] curiosity and the desire to support research. [39] In one study, a self-stereotyped belief of being a “natural athlete” facilitated participation. [31] In another study, use of digital health trackers were facilitators as they increased self-awareness of activity levels, however, difficulties with the technology was a barrier. [50] A lack of self-motivation was a barrier in all geographic locations [32,33,36,44,49,50], while a lack of confidence around trying something new and general attitudes towards physical activity were barriers in only one urban and rural/regional study [50]. The practical need to be active in instances, such as walking for transport, was a facilitator reported in different geographic locations [33,51], while one study reported the integration of physical activity into daily life and culture as a facilitator [31]. Lack of transport and financial constraints were barriers in numerous studies [31,32,33,34,39,41,42,46,47,48,54]. Resource constraints were a barrier in two studies [18,38] and unemployment was a barrier in one rural/regional study [47]. Lack of time was a barrier across various locations, and was often associated with work, study, or other commitments [33,34,39,47,49,50]. Other individual circumstance barriers included other travel commitments in rural/regional areas [47], and major life events in an urban study. [39]

Action Statement 2: Promoting the holistic health and personal benefits of physical activity, whilst also addressing the challenges of those facing specific physical or health challenges, may facilitate participation.

Subcategories: Expected and realized personal benefits, health goals and issues and overcoming specific challenges 

A range of health-related motivators were described as facilitating participation, as was having expectations of and realizing various benefits from physical activity. However, a number of health or physical issues functioned as barriers, requiring strategies in order to overcome them. Expecting benefits from physical activity was a facilitator for participating in physical activity across various geographic locations [33,49]. The realized benefits that facilitated participation included enjoyment [36,38,39,53], improved health in urban studies [39,44,53], and having a purpose in remote studies [35,54]. Health-related facilitators across locations included personal motivations for fitness [36,38,47,49] and having knowledge of First Nations health issues and risk factors [36,41]. Across various locations, people with previous experiences of illness or injury, or current health problems, were motivated to participate in physical activity, often in order to mitigate disease progression [35,36,39,41,47,51]. Other facilitators included a desire for mental wellbeing [36,38] and, in remote studies, a desire for weight control [36,51] and feeling happy and strong [35]. However, injury or illness was a barrier in all locations [33,34,35,36,37,42,45,49], and poor mental health was a barrier in one study [35]. 

Disability was a physical individual barrier in several locations [35,37,42]. In remote areas, perceived age and weight constraints were a barrier, as was changing long-term behavior in those with or at risk of type 2 diabetes [36,52]. However, certain strategies were facilitators, such as making slow, small behavioral changes [52] and using assistive devices and modifying activities in instances of disability [35].

### 3.6. Interpersonal Facilitators and Barriers 

Statement 3: Recognizing the importance of family and cultural connections and providing opportunities for positive connections for Aboriginal and Torres Strait Islander people with family, peers and broader networks may facilitate physical activity participation. 

Subcategories: Impacting and being impacted by other people, family and peers and social network. 

The importance and influence of family, friends, community members and role models regarding physical activity participation was reflected in many of the facilitators and barriers. Fostering interpersonal connections while addressing broader social issues, such as racism or conflict, may play an important role in facilitating physical activity participation. 

Across geographic locations, the influence of role models was an interpersonal facilitator for participation in physical activity [34,36,48], as was the influence of family and friends [36,44,46,48]. In one study the importance of having a role model or program leader of the same gender was highlighted as a facilitator [47]. Wanting to be a role model for children and others was a facilitator [30,31,32,46]. In one study, the influence of having health information shared with an individual on social media was a facilitator [48]. However, in one regional/rural study stigma around physical activity had a negative influence on participation [47]. Experiences of racism was a barrier in all geographic locations, as was public judgement in one urban study [31,33,38,41,42]. However, one urban study described wanting to challenge racism as a facilitator [38].

Support from family, including provision of material and instrumental assistance, was a facilitator across locations [35,39,46,47,52]. Inclusion of family members in physical activities was also a facilitator [38,42,47]. On the inverse, a remote study described a lack of family inclusion as a barrier [54], while a rural study described a lack of family support as a barrier [47]. Across locations, family commitments, including caring for children, were a common barrier [31,33,39,42,43,45,46,49,50], as was prioritizing children’s participation in sport and physical activity [33]. For women, gendered roles and responsibilities, such as housework, were barriers [32,43]. In urban studies, family and community conflict were barriers to participation [34,38].

Across various locations, relationships with program staff that were supportive, respectful, or encouraging were facilitators [32,36,45,47,53], as was receiving peer support [38,47,52,53]. The opportunity for social connections was a commonly reported facilitator [31,32,34,38,39,54]. Additionally, the opportunity to participate in group activities or have exercise companions were facilitators and, in urban studies, the participation of others from a participant’s community network and a supportive, positive group environment were facilitators [42,45,46,53]. For some urban women, a women-only group was a facilitator [29,30]. 

Shame and embarrassment were barriers across various geographic locations, but were noted as becoming less so in some communities [33,35,36,43,47]. Lack of support from peers was a barrier [32,47]. Concern for the safety of older people was barrier to going On-Country in remote areas [18], while a non-Indigenous group atmosphere was a barrier in a rural/regional study [46]. In urban locations, conflict with program staff was a barrier [34], and peer rivalry was a barrier for sport participation [38]. In one individualized intervention, a lack of social interaction was a barrier to participation [50]. Finally, across locations, cultural obligations, including funerals, was a barrier [18,45,47].

### 3.7. Community and Environment Facilitators and Barriers 

Statement 4: Respecting connections to culture and supporting communities to be supportive, safe, and well-resourced may facilitate participation in physical activity and sport.

Subcategories: Community context, safety and resources, community connection and connecting with culture. 

Community-level facilitators and barriers suggest that adequate infrastructure and neighborhood safety are important factors influencing physical activity. Community relationships also play an important role that can help or hinder physical activity participation. Connecting to culture and access to culturally safe places and activities may be an important factor for Aboriginal and Torres Strait Islander people for engaging in physical activity.

Access to equipment in an urban study [53] and appealing and varied locations for exercise in a remote study [36] were community and environmental factors that facilitated participation. However, across various locations a lack of available services and physical activity opportunities was reported [31,33,42]. Similarly, across various locations unsafe or inadequate infrastructure [31,32,36,46] and general safety concerns [36,42,46] were a barrier, the latter were particularly noted among women [36,42,46]. A lack of access to facilities was a barrier in a rural study [47]. In remote locations, dangerous dogs were a barrier [36], and unappealing outstations and the distractions of community life were barriers to participating in On-Country physical activity [18]. Urban traffic and urban living itself were barriers [39,42]. Additionally, local weather and climate were barriers (e.g., heat or rain) [18,32].

Community-wide healthy lifestyle changes or attitudes were a facilitator to individual physical activity across various locations [36,48]. In an urban study [38] and a remote study [36], the ability to connect with community was a facilitator. However, the pressure of community expectations to excel was a barrier for sport participation [38]. High death rates had wide community impacts and were a barrier [34]. In a rural/regional study, unfriendly neighborhood environments was a barrier [46], while in a remote study, distractions of community life was a barrier [18].

Participating in cultural activities was a facilitator across various geographic locations [35,40,51]. Having access to a culturally appropriate and safe environment [31,39,49], or an Aboriginal and Torres Strait Islander specific facility or activity [41] were also facilitators. In an urban study, a sense of community history and pride associated with sport were facilitators [38]. Culturally inappropriate activities and a lack of cultural inclusiveness were barriers to participation across locations [33,51]. Participants in urban areas reported disrupted connection with culture and land as barriers [41].

### 3.8. Policy and Program Facilitators and Barriers 

Statement 5: Physical activity and sport programs should be sustainably resourced and receptive to participants’ needs and expectations to facilitate participation. 

Subcategories: Program delivery and external support.

Physical activity and sport programs can facilitate attendance with sufficient and sustainable funding and staffing, and with program designs that accommodate the needs and expectations of Aboriginal and Torres Strait Islander people. Many features of physical activity programs and interventions were considered to be facilitators, including being cost-free and structured, and in urban studies, these included the provision of transport, childcare and professionally delivered and well-organized programs [39,45,47,53], although flexibility for participants was also a facilitator [45]. Other urban program facilitators were convenient times and locations, variety in activities, meeting needs and expectations and being connected with an Aboriginal community-controlled health organization [39,45,53]. Having a positive program experience facilitated ongoing participation [39].

**Table 3 ijerph-18-09893-t003:** Facilitators and barriers to physical activity and sport participation experienced by Aboriginal and Torres Strait Islander adults, by geographic location.

Socio-Ecological Level	Urban		Regional/Rural		Remote	
	Facilitators	Barriers	Facilitators	Barriers	Facilitators	Barriers
Individual	Health problems/desire to prevent disease progression [39] Expected benefit [33,42] Practical need, i.e., active transport [33] Desire to improve fitness [38,49] Enjoyment [38,39,53] Personal experience of injury or illness [41] Knowledge of First Nations health issues and risk factors [41] Digital health trackers and increased self-awareness of activity levels [50] Relieve stress, mental wellbeing [38] Self-motivation [32,39,49] Curiosity [39] Desire to support research [39] Poor body image perceptions [32] Feeling healthier/improved health [39,44,53] Learning new health information [53] Self-stereotype as natural athlete [31] Physical activity integrated into culture and daily life [31]	Lack of access to transport and logistical difficulty [33,34,39,41] Lack of self-motivation [32,33,49,50] Financial constraints [31,32,33,39,41,42] Lack of time [33,49,50] Injury or illness [33,34,42,45,49] Disability [42] Work commitments [33,34,39,50] Lack of resources [38] Study commitments [39,50] Challenges with digital health tracker technology [50] Other commitments [39] Major life events [39] Lack of confidence to try something new [44] General attitude to health and exercise [44] Pain from exercising the previous day [49]	Health problems/desire to prevent disease progression [47] Expected benefit [33] Practical need, i.e., active transport [33] Desire to improve fitness [47] Personal experience of injury or illness [41] Knowledge of First Nations health issues and risk factors [41] Digital health trackers and increased self-awareness of activity levels [50]	Lack of access to transport and logistical difficulty [33,41,46,47] Lack of self-motivation [33,50] Financial constraints [33,41,47] Lack of time [33,50] Injury or illness [33,37] Disability [37] Work commitments [33,47,50] Unemployment [47] Travelling for other reasons [47] Menstruation [47] Study commitments [50] Challenges with digital health tracker technology [50]	Health problems/desire to prevent disease progression [35,36,51] Expected benefit [33] Practical need, i.e., active transport [33,51] Desire to improve fitness [36] Enjoyment [36] Personal experience of injury or illness [41] Knowledge of First Nations health issues and risk factors [36,41] Weight control [36,51] Making small, slow behaviour changes [52] Relieve stress, mental wellbeing [36] Having a purpose [35,54] Assistive devices and home modifications [35] Modifying activities [35] Feeling internally strong/happy [35] Self-stereotype as natural athlete [31] Physical activity integrated into culture and daily life [31]	Lack of access to transport and logistical difficulty [31,33,54] Lack of self-motivation [33,36] Financial constraints [31,33,41] Lack of time [33] Injury or illness [33,37] Disability [35,37] Work commitments [33] Poor mental health [35] Lack of resources [18] Difficulty changing long-term behaviour [52] Perceived age or weight constraints [36]
Interpersonal	Peer support [38,53] Family support including material/instrumental support [39] Influence of role models [34,48] Influence of family [44,48] Influence of friends [48] Program staff support, respect, encouragement [39,45] Role-modelling for children [32] Inclusion of families in activities [38,42] Competition [38,50] Information sharing on social media [48] Group activities and exercise companions [42,45,53] Social connections [31,32,34,38,39] Challenging racism [38] Participation of others from community network [53] Positive/supportive group atmosphere [39,45] Women-only groups [29]	Cultural obligations including funerals [45] Racism [31,33,38,41] Family commitments including caring for children [31,33,39,42,45,49,50] Shame and embarrassment [33] Prioritising children’s participation [33] Gender roles and responsibilities [32] Lack of peer support [32] Peer rivalry [38] Lack of social interaction [50] Community/family conflict [34,38] Conflict with program staff [34] Public judgement [42]	Peer support [47] Family support including material/instrumental support [46,47] Influence of role models [48] Influence of family [46,48] Influence of friends [48] Program staff support, respect, encouragement [47] Role-modelling for children [46] Inclusion of families in activities [47] Competition [50] Information sharing on social media [48] Group activities and exercise companions [46] Role model/program leader of same gender [47]	Cultural obligations including funerals and Sorry Business [47] Racism [33,41] Family commitments including caring for children [33,46,50] Shame and embarrassment [33,47] Prioritising children’s participation [33] Lack of family support [47] Lack of peer support [47] Stigma around physical activity [47] Lack of social interaction [50] Non-Indigenous group atmosphere [46]	Peer support [52] Family support including material/instrumental support [35,52] Influence of role models [36] Influence of family [36] Influence of friends [36] Program staff support, respect, encouragement [36] Role-modelling for others [30,31] Social connections [31,54]	Cultural obligations including funerals [18] Racism [31,33,41] Family commitments including caring for children [31,33,43] Shame and embarrassment [33,35,36,43] Prioritising children’s participation [33] Gender roles and responsibilities [43] Families not included [54] Safety concerns for elderly [18]
Community	Community health behaviour, attitudes [48] Cultural activities [40] Culturally appropriate/ culturally safe environment [31,39,49] Aboriginal and Torres Strait Islander specific facility or activity [41] History and pride [38] Access to equipment [53] Community connections [38]	Culturally inappropriate activities/lack of cultural inclusiveness [33] Lack of available services/physical activity opportunities [31,33,42] General safety concerns [42] Unsafe or inadequate infrastructure [31,32] Weather and climate [32] Traffic [39] Urban setting [42] Disrupted connection with culture and land [41] Community expectations [38] High community death rates [34]	Community health behaviour, attitudes [48] Cultural activities [40] Aboriginal and Torres Strait Islander specific facility or activity [41]	Culturally inappropriate activities/lack of cultural inclusiveness [33] Lack of available services/physical activity opportunities [33] General safety concerns [46] Unsafe or inadequate infrastructure [46] Unfriendly and uncomfortable neighbourhood [46] Lack of access to facilities [47] Disrupted connection with culture and land [29]	Community health behaviour, attitudes [36] Cultural activities [35,51] Culturally appropriate/culturally safe environment [31] Appealing and varied locations for activity [36] Community connections [36]	Culturally inappropriate activities/lack of cultural inclusiveness [33,51] Lack of available services/physical activity opportunities [31,33] General safety concerns [36] Unsafe or inadequate infrastructure [31,36] Weather and climate [18] Dangerous dogs [36] Unappealing outstations [18] Distractions of community life [18]
Program and Policy	Free program [31,39,45] Supportive employers [34] Provision of transport [45,53] Structured program [53] Positive program experience [39] Convenient times and location [39] Provision of childcare [39] Program meets needs and expectations [39] Professionalism/ well-organised program [39] Program connected to local Aboriginal community-controlled health organisation [45] Flexibility [45] Variety of exercises in program [53]	Inconvenient program location [39] Program different to expectations, mismatched with fitness level [39] Lack of sustainable, local physical activity programs [42] Insufficient number of programs and locations [44] Lack of motivation, confidence of initiative around chronic disease self-management [44] Lack of knowledge about programs [44]	Free program [47]	Session times and frequency [47]	Support from services [35]	No one to run program [54] Session times and frequency [54] Loss of program funding [54] Lack of sustainable, local physical activity programs [43] Reliance on welfare [43]

In a rural/regional study [47] and a remote study [54], finding feasible session times and frequency were a barrier to participation and, in a remote setting, loss of program funding and having no one to run programs were barriers [54]. However, an urban and a remote study also cited lack of sustainable local physical activity initiatives as a barrier [42,43]. Other urban program barriers were a lack of knowledge about the program [44], a mismatch of participant expectations and fitness level with the program [39], inconvenient program location [39] and an insufficient number of programs and locations [44]. Reliance on welfare was a policy barrier to participation in physical activity in one study, as past policies had an ongoing disempowering effect on the community, leading them to depend on external agencies to provide participation opportunities [43]. Organizations could play a role in facilitating physical activity by supporting employees to attend programs [34].

## 4. Discussion

This mixed methods systematic review synthesized 27 studies, finding many diverse facilitators and barriers to physical activity and sport participation that are experienced by Aboriginal and Torres Strait Islander adults. Most facilitators and barriers identified were individual and interpersonal, as categorized by the socio-ecological model of health [20]. There were similar numbers of facilitators and barriers identified at each level of the socio-ecological model, except for at the community level where more barriers than facilitators were identified. Most studies were qualitative and used similar methods, generally using individual or group interviews or the First Nations methods of Yarning and Dadirri. Yarning involves a conversational process of knowledge and story sharing [55] and Dadirri is a way of life encapsulating deep listening, sharing and trust [56]. The use of these methods was appropriate to elicit in-depth and culturally relevant complexities surrounding the unique facilitators and barriers to physical activity and sport participation experienced by Aboriginal and Torres Strait Islander adults. First Nations methods and worldviews were also evident in the included mixed-method and quantitative studies and their collective synthesis gives rich data on lived experiences.

The review found that the personal attitudes and life circumstances of Aboriginal and Torres Strait Islander people should be considered for effectively identifying and addressing opportunities for physical activity and sport, and potential barriers. This finding is relevant at multiple levels of the socioecological model, where personal attitudes pertain to the individual and inter-personal level and the upstream levels of community, environment, policy and program that impact life circumstances. Given the complex historical factors that have shaped Aboriginal and Torres Strait Islander health and wellbeing and life experiences today, strategies to encourage physical activity and sport participation and to reduce barriers to participate must be relevant to contemporary experiences [2].

Many personal facilitators and barriers were associated with concepts of health and wellbeing highlighting, as summarized in Action Statement 2, that promoting the health and personal physical activity benefits whilst addressing the challenges of those facing specific physical or health challenges is crucial. The desire to improve health and fitness were facilitators for physical activity, especially if someone had experienced illness or injury or was living with a chronic disease. Two studies [41,47] explain this through the health belief model [57], where perceived health threats and cues for action can facilitate behavior change, although the cultural relevance of the model is unknown. Interrelationships were also seen between physical activity participation and mental health, with feeling happy and strong and wanting stress relief facilitating participation, while poor mental health was a barrier to participation. This highlights the role that sport and physical activity can play in improving wellbeing, given the historical factors and intergenerational trauma that have led to a high burden of mental illness and stress for Aboriginal and Torres Strait Islander people [58]. The positive impact of sport and physical activity on social and emotional wellbeing experienced by Aboriginal and Torres Strait Islander people is suggested [10], but greater empirical evidence is required across the life course.

Barriers such as a lack of access to transport, financial constraints and unemployment may interrelate with lower socioeconomic status. Again, due to the ongoing impacts of European invasion and racial discrimination, Aboriginal and Torres Strait Islander adults are disproportionally within the lowest income quintile, with implications for accessing health-related goods and services [6]. A recent longitudinal study found a statistically significant association between area socioeconomic status and participation in physical activity among Australian adults [59]. Among Aboriginal and Torres Strait Islander adults in NSW, high physical activity levels were associated with a higher socioeconomic position [60]. The present review also demonstrated several program factors that facilitate participation and overcome these barriers, including the provision of transport and programs at no cost to participants, which suggest important elements for program modification and future design [61].

The importance of social and family connections was very apparent, with opportunities for these connections facilitating physical activity and sport participation, whilst family and cultural obligations may be barriers to participation. Further, respecting connections to culture and supporting communities to be supportive, safe, and well-resourced may facilitate physical activity and sport participation. These findings highlight the central roles of family, community, kinship and Country for Aboriginal and Torres Strait Islander culture, society [62], health and wellbeing [63]. This has significant implications for physical activity program providers, who can facilitate participation by ensuring family and community inclusion and togetherness. Similarly, integrating physical activity into daily life and culture is an important facilitator and should be supported. There is emerging evidence that cultural and Country-based activities that include physical activity improve health and wellbeing [64]. Additionally, having role models and desiring to be a role model for others was a facilitator for physical activity. This was also reflected in the review of child and adolescent barriers and facilitators, [20] where family participation and role modelling had flow-on effects for physical activity in young Aboriginal and Torres Strait Islander people, further supporting the need for inclusive, culturally relevant, and family-oriented physical activity initiatives.

Experiences of racism were a barrier for physical activity and sport participation. The need for culturally safe facilities, appropriate activities and inclusive, respectful staff were all described as facilitators and provide important insights for program providers. This corroborates the proposed roles of Aboriginal and Torres Strait Islander sports teams and programs in providing opportunities to assert identity and increase the visibility of success, as well as being hubs for cultural safety and community connection [65]. In the present review, self-stereotyping around perceptions of Aboriginal and Torres Strait Islander people’s “natural ability” in sports was a facilitator to participation. This stereotype has been upheld previously and, while self-belief may enable participation, the stereotype has been critiqued as undermining other capabilities of Aboriginal and Torres Strait Islander people, such as intelligence, and limiting avenues for success in other fields [66].

Transport and logistical issues were barriers across various geographic contexts, but with some variation according to remoteness. A lack of public transport was a reported issue in rural and remote locations, with remote participants also facing issues of high fuel costs and difficulties related to travelling between communities. In urban locations, reliability and affordability of transport was an issue, and people, at times, needed to walk significant distances to access facilities. Previous research with Aboriginal people aged 45 years and over also indicated the importance of neighborhood characteristics, such as a lack of public transport, where supportive neighborhood environments were associated with higher physical activity participation [60]. In this review, traffic had negative effects on motivation in urban areas, and urban dwelling and disrupted connections with Country and culture emerged as barriers to participation. The cultural appropriateness of activities, like running and swimming, varied significantly across the studies and cultural groups, demonstrating the importance of local context and using collaborative approaches for programs and environmental provision for community-based activities and facilities.

Several barriers specific to women were identified, including child-raising and household obligations and environmental safety concerns. Women prioritized the physical activity of their partners who were the main income earners, reflecting the roles Aboriginal women have played as “enablers” for the participation of others. Shame, a complex concept that can involve knowledge and emotions arising from being singled out for negative and positive reasons [67], was a barrier for women, but this was also challenged as an “old-fashioned” notion [43]. Having women-only activities was a strong facilitator of sport and physical activity participation. Understandings of physical activity were also noted to be different between genders, with women including not just sport but also household work and other activities as physical activity. Understanding the physical activity and sport experiences of Aboriginal and Torres Strait Islander women is particularly important, given gender disparities demonstrated in demographic studies, particularly among adolescent females [68].

At the policy and program level, our findings revealed that physical activity and sport programs should be sustainably resourced and receptive to participants’ needs and expectations to facilitate participation. Physical activity and sport program provision for Aboriginal and Torres Strait Islander people is common but program impact could be better determined [61]. Several studies were program evaluations, yielding facilitators that can be considered in program design and adaptation, including addressing affordability, transport and incorporating variety and flexibility. Funding and long-term sustainability were identified as program barriers, with “one-off” events viewed as insufficient to have an impact on participation. Thus, true community engagement will play a critical role in facilitating sustainable physical activity and sport program offerings for Aboriginal and Torres Strait Islander people; recommendations that are commonly made but less often realized. Such engagement includes community co-design, leadership and governance of programs that are inherent, as per the Aboriginal and Torres Strait Islander QAT, along with political support and sustainable funding sources.

In this systematic review we used this QAT to assess the included studies. This recently developed tool is the first to assess research quality from an Aboriginal and Torres Strait Islander perspective [28]. The areas in which the most studies performed well were using a strengths-based approach, using First Nations research paradigms and strengthening capacity for Aboriginal and Torres Strait Islander people. However, very few studies reported Aboriginal and Torres Strait Islander research governance, community control over collection and management of research materials, or negotiated agreements of participant and community rights of access to existing intellectual and cultural property. No studies indicated a negotiated agreement had been made to protect Aboriginal and Torres Strait Islander peoples’ ownership of the intellectual and cultural property created through the research. Overall, there was a lack of clear reporting of steps taken to ensure the research aligned with community interests, although some studies had included more details in associated papers, such as protocols. The QAT provides an opportunity for researchers to improve the conduct and report of research by using the checklist as a guide for ethical research practices and by including, in publications, evidence of the actions taken to ensure that the research reflects the interests of Aboriginal and Torres Strait Islander communities.

Utilizing a mixed methods review methodology is a strength of this paper, as it allows for the identification and synthesis of concepts across a variety of study types [69]. Categorizing facilitators and barriers with a socio-ecological model is another strength, as it allows for understanding of interacting influences at the personal, social, environmental and political levels [70]. The review also complements a recent Aboriginal and Torres Strait Islander child physical activity and sport review, providing comprehensive evidence synthesis across the life course [20]. Furthermore, the review comprehensively extracted, synthesized and reflected on studies examining the complex terms “barrier” and “facilitator” in the context of Aboriginal and Torres Strait Islander experiences of physical activity and sport [11]. There are some limitations: only two longitudinal studies were identified, and, thus, understandings of changes to facilitators and barriers over time is very limited. While most studies used purposive or convenience sampling, this is appropriate for qualitative studies [71] and the importance of the local cultural context and diversity can be emphasized. Nevertheless, caution must be taken when generalizing findings across Aboriginal and Torres Strait Islander populations, regardless of sampling methods, given the cultural and contextual diversity in these populations [5]. The incorporation of two diverse quality assessment tools [27,28] allows for better understanding of the strength and cultural relevance of evidence and indicates how research can be better articulated in order to reflect on and respond to community priorities [27].

Finally, facilitating participation in physical activity is likely to have a positive effect on Aboriginal and Torres Strait Islander health. [72] However, a holistic approach is required for improving health outcomes, given the many social and cultural determinants which impact health [4]. Addressing these determinants is likely to address many of the barriers to physical activity that are identified in this review, including issues of affordability and access. Given the multitude of facilitators and barriers influencing participation in physical activity and sport for Aboriginal and Torres Strait Islander people, future strategies should strongly consider, plan for, and evaluate the impact of facilitators and barriers for participation.

## 5. Conclusions

This mixed methods systematic review has identified multiple, complex facilitators and barriers experienced by Aboriginal and Torres Strait Islander adults participating in physical activity and sport across different contexts and locations. The implication of these findings is for programs and decision makers to address these factors in collaboration with Aboriginal and Torres Strait Islander communities and with consideration of the local community context, to facilitate increased participation. The five action statements generated through the synthesis of the mixed method studies within this review give clear practical guidance for future program and planning, as well as for improving current program delivery. More broadly, public policy that recognizes and seeks to address the social determinants of health affecting Aboriginal and Torres Strait Islander people will help to address some of the upstream barriers to participation and create a supportive environment to facilitate participation.

## Figures and Tables

**Figure 1 ijerph-18-09893-f001:**
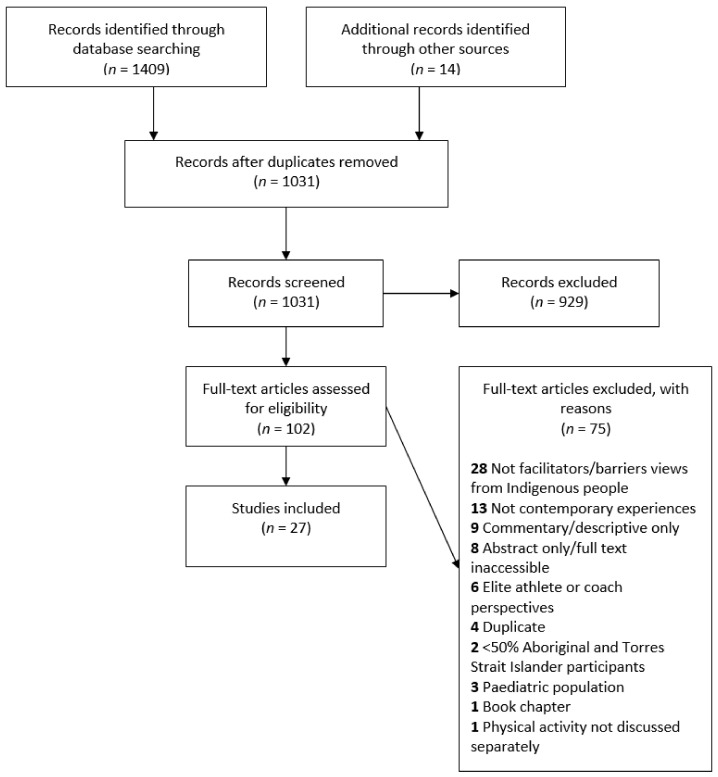
PRISMA flowchart for study identification and inclusion.

**Table 2 ijerph-18-09893-t002:** Facilitators and barriers to physical activity and sport participation experienced by Aboriginal and Torres Strait Islander adults.

Socio-Ecological Level	Facilitators	Barriers
Individual	*Self-beliefs and attitudes* Poor body image perceptions [32] Self-motivation [32,39,49] Curiosity [39] Desire to support research [39] Self-stereotype as natural athlete [31] Digital health trackers and increased self-awareness of activity levels [50] *Physical activity aligned with daily life* Physical activity integrated into culture and daily life [31] Practical need, i.e., active transport [33,51] *Expected and realised benefits* Expected benefit [33,49] Enjoyment [36,38,39,53] Having a purpose [35,54] Feeling healthier/improved health [39,44,53] *Health goals and issues * Desire to improve fitness [36,38,47,49] Relieve stress, mental wellbeing [36,38] Knowledge of Aboriginal and Torres Strait Islander health issues and risk factors [36,41] Personal experience of injury or illness [41] Learning new health information [53] Health problems/desire to prevent disease progression [35,36,39,47,51] Weight control [36,51] Feeling internally strong/happy [35] *Overcoming specific challenges* Making small, slow behaviour changes [52] Assistive devices and home modifications [35] Modifying activities [35]	*Self-beliefs and attitudes* Lack of self-motivation [32,33,36,49,50] Lack of confidence to try something new [44] General attitude to health and exercise [44] Challenges with digital health tracker technology [50] *Personal circumstances* Lack of access to transport and logistical difficulty [31,33,34,39,41,46,47,54] Financial constraints [31,32,33,39,41,42,46,47] Unemployment [47] Lack of time [33,49,50] Lack of resources [18,38] Work commitments [33,34,39,47,50] Study commitments [39,50] Other commitments [39] Travelling for other reasons [47] Major life events [39] *Health goals and issues* Injury or illness [33,34,37,42,45,49] Poor mental health [35] Menstruation [47] Pain from exercising the previous day [49] *Other specific challenges* Difficulty changing long-term behaviour [52] Disability [35,37,42] Perceived age or weight constraints [36]
Interpersonal	*Having an impact and being impacted by others* Influence of role models [34,36,48] Role model/program leader of same gender [47] Influence of family [36,44,46,48] Influence of friends [36,48] Role-modelling for children [32,46] Role-modelling for others [30,31] Information sharing on social media [48] Challenging racism [38] *Family* Inclusion of families in activities [38,42,47] Family support including material/instrumental support [35,39,46,47,52] *Peers and social networks* Social connections [31,32,34,38,39,54] Peer support [38,47,52,53] Competition [38,50] Program staff support, respect, encouragement [36,39,45,47,53] Group activities and exercise companions [42,45,46,53] Participation of others from community network [53] Positive/supportive group atmosphere [39,45] Women-only groups [29,30]	*Having an impact and being impacted by others* Racism [31,33,38,41] Public judgement [42] Stigma around physical activity [47] *Family* Families not included [54] Gender roles and responsibilities [32,43] Lack of family support [47] Family commitments including caring for children [31,33,39,42,43,45,46,49,50] Prioritising children’s participation [33] Community/family conflict [34,38] *Peers and social network* Lack of peer support [32,47] Shame and embarrassment [33,35,36,43,47] Peer rivalry [38] Safety concerns for elderly [18] Conflict with program staff [34] Lack of social interaction [50] Non-Indigenous group atmosphere [46] Cultural obligations, including funerals and Sorry Business [18,45,47]
Community/Environment	*Community context, safety and resources* Access to equipment [53] Appealing and varied locations for activity [36] *Community relationships* Community health behaviour, attitudes and initiatives [36,48] Community connections [36,38] History and pride [38] *Connecting with culture* Culturally appropriate/culturally safe environment [31,39,49] Aboriginal and Torres Strait Islander specific facility or activity [41] Cultural activities [35,40,51]	*Community context, safety and resources* Weather and climate [18,32] Unsafe or inadequate infrastructure [31,32,36,46] General safety concerns [36,42,46] Dangerous dogs [36] Lack of available services/physical activity opportunities [31,33,42] Lack of access to facilities [47] Unappealing outstations [18] Traffic [39] Urban setting [42] *Community relationships* Unfriendly and uncomfortable neighbourhood [46] Community expectations [38] High community death rates [34] Distractions of community life [18] *Connecting with culture* Culturally inappropriate activities/lack of cultural inclusiveness [33,51] Disrupted connection with culture and land [41]
Program/Policy	*Program delivery* Provision of transport [45,53] Structured program [53] Free program [39,45,47] Positive program experience [39] Convenient times and location [39] Provision of childcare [39] Program meets needs and expectations [39] Professionalism/well-organised program [39] Program connected to local Aboriginal community-controlled health organisation [45] Flexibility [45] Variety of exercises in program [53] *External support* Supportive employers [34] Support from services [35]	*Program delivery* No one to run program [54] Session times and frequency [47,54] Loss of program funding [54] Inconvenient program location [39] Program different to expectations, mismatched with fitness level [39] Lack of sustainable, local physical activity programs [42,43] Insufficient number of programs and locations [44] Lack of motivation, confidence or initiative around chronic disease self-management [44] Lack of knowledge about programs [44] Reliance on welfare [43]

## Data Availability

Not applicable.

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
