# Peer review of "Facilitators and Barriers to Physical Activity and Sport Participation Experienced by Aboriginal and Torres Strait Islander Adults: A Mixed Method Review"

_ijerph, 2021, doi:10.3390/ijerph18189893_

Round 1

Reviewer 1 Report

The manuscript is a systematic review using mixed methods and guided by the socio-ecological framework to identify facilitators and barriers to adult physical activity and sport participation among Aboriginal and Torres Strait Islander adults. The topic is salient and of interest to the journal readers. The authors performed a thorough review. The structure and content of the manuscript are well organized. The work, in general, is fundamentally sound and comprehensive. I only have a couple of suggestions basically on the paper appearance.

  1. The last paragraph of the Introduction section was swift statements to introduce the review. Perhaps, the authors can emphasize that review based on the socio-ecological framework.
  2. In figure 1, the number of studies included in the review was 26; however, the manuscript text indicated throughout the paper that they have a final number of 27.
  3. Table 1 and Supplements may be best using landscape orientation rather than portrait.

Author Response

Reviewer 1

The manuscript is a systematic review using mixed methods and guided by the socio-ecological framework to identify facilitators and barriers to adult physical activity and sport participation among Aboriginal and Torres Strait Islander adults. The topic is salient and of interest to the journal readers. The authors performed a thorough review. The structure and content of the manuscript are well organized. The work, in general, is fundamentally sound and comprehensive. I only have a couple of suggestions basically on the paper appearance.

1. The last paragraph of the Introduction section was swift statements to introduce the review. Perhaps, the authors can emphasize that review based on the socio-ecological framework.

 Thank you for your helpful review and suggestions.

We have emphasised that the review uses a socio-ecological framework on lines 117-118.

2. In figure 1, the number of studies included in the review was 26; however, the manuscript text indicated throughout the paper that they have a final number of 27.

We have corrected this discrepancy; it is 27 studies.  

3. Table 1 and Supplements may be best using landscape orientation rather than portrait.

Table 1 and the supplements are in landscape orientation

Reviewer 2 Report

This manuscript entitled "Facilitators and barriers to physical activity and sport participation experienced by Aboriginal and Torres Strait Islander adults: a mixed method review" aimed to  synthetize existing evidence on facilitators and barriers for physical activity participation experienced by Aboriginal and Torres Strait Islander adults in Australia.

The manuscript is very interesting. However, some relevant issues should be addressed by the authors:

RESULTS SECTION

  • Figure 1 and topic 3.1 (lines 187-191):   Please, cheack all numbers regarding the flow of included articles. The numbers are different between text and Figure 1.
  • Table 3: This table is quite difficult  to understand. Please, improve the format, design, and flow.

DISCUTION SECTION

  • Please, include clearly the limitations.

Author Response

Reviewer 2

Author response

This manuscript entitled "Facilitators and barriers to physical activity and sport participation experienced by Aboriginal and Torres Strait Islander adults: a mixed method review" aimed to  synthetize existing evidence on facilitators and barriers for physical activity participation experienced by Aboriginal and Torres Strait Islander adults in Australia.

The manuscript is very interesting. However, some relevant issues should be addressed by the authors:

RESULTS SECTION

1. Figure 1 and topic 3.1 (lines 187-191):   Please, check all numbers regarding the flow of included articles. The numbers are different between text and Figure 1.

Thank you for your helpful review and suggestions.

We have corrected this discrepancy

2. Table 3: This table is quite difficult  to understand. Please, improve the format, design, and flow.

We have formatted the table and its layout to be clearer.

3. DISCUTION SECTION

Please, include clearly the limitations.

We have emphasised the limitations in lines 600 - 606.

Reviewer 3 Report

This paper provides a mixed-methods review of the literature on barriers and facilitators to physical activity amongst Aboriginal and Torres Strait Islander adults. Overall it is clearly written, and the content is of importance given the ongoing health inequity. It is comprehensive and findings are well structured. Thank you for the opportunity to review this interesting paper. I have no methodological concerns. Some small comments/suggested edits provided below.

Intro

Well-presented and generally clear.

Line 50-53 – this sentence is a bit unclear. Perhaps it needs a comma after ‘discrimination’? Or perhaps ‘and racism’ becomes ‘whilst racism’? Assuming the statement is that racism remains an ongoing structural determinant of health

Methods

Paragraph 3 – inclusion/exclusion criteria – this was a bit difficult to follow. Consider grouping all inclusion criteria together and all exclusions together. Or summarising in a table.

Line 163 – check the expression here, the sentence is problematic (perhaps from editing process) and needs a rewrite: “Full texts were of studies included after abstract screening were uploaded….”

Line 168 and 169 – you treat data as singular then as plural –please be consistent (and preferably treat as plural)

Results

Lines 209-2010 – the ‘and’ before ‘three in Western Australia’ is not needed, and ‘in two Victoria’ needs a quick edit

Line 225 – can you provide more detail on this? E.g., – do you mean they developed theoretical framework or used ones already developed and/or well-established in health behaviour literature? If they used previously developed theoretical frameworks can you provide detail as to which? -assuming some studies used the same or similar frameworks. It may be interesting to know which were most commonly used. Or were these all totally different frameworks?

Table 1 – this is difficult to read given its current presentation with words split across multiple lines.

Also – in Theoretical Framework – there are both epistemologies and theoretical frameworks in here – I recommend splitting these out so its clear what previously developed theoretical frameworks (e.g., behavioural theories) are being used and, separately, what epistemologies underpin the qual approaches

Table 2 – ‘digital health tackers’ should be ‘digital health trackers’

Section 3.7 – I recommend removing this section text given its brevity but incorporating the statement earlier to highlight the inclusion of geographic information in the previous section and refer the reader to Table 3.

Table 3 – formatting issues – column headings – Facilitators and Barriers are not aligned with different columns (i.e., both kind of aligned with the same columns). Is it possible to incorporate this info into table 2? Perhaps by including identifiers next to the statements within the table – e.g., U – urban; RR – regional/rural; R remote – or some similar code so that you can include the key letters next to the statements similar to what you have done with the reference numbers. It might make it easier for the reader to see al of the information, especially given the difficulties with so many columns in table 3 making the text difficult to read easily.

Discussion

Good.

Author Response

Reviewer 3

Author response

This paper provides a mixed-methods review of the literature on barriers and facilitators to physical activity amongst Aboriginal and Torres Strait Islander adults. Overall it is clearly written, and the content is of importance given the ongoing health inequity. It is comprehensive and findings are well structured. Thank you for the opportunity to review this interesting paper. I have no methodological concerns. Some small comments/suggested edits provided below.

Intro

Well-presented and generally clear.

Line 50-53 – this sentence is a bit unclear. Perhaps it needs a comma after ‘discrimination’? Or perhaps ‘and racism’ becomes ‘whilst racism’? Assuming the statement is that racism remains an ongoing structural determinant of health

Thank you for your review.

We have rewritten this sentence to read more clearly “These nations share common histories of marginalizing First Nations peoples through practices such as genocide, dispossession and exclusion, whilst discrimination and racism remain ongoing structural determinants of health.”

Methods

Paragraph 3 – inclusion/exclusion criteria – this was a bit difficult to follow. Consider grouping all inclusion criteria together and all exclusions together. Or summarising in a table.

We have edited the criteria to focus on the inclusion criteria first (lines 141-158), followed by exclusion criteria (lines 159 – 163)

Line 163 – check the expression here, the sentence is problematic (perhaps from editing process) and needs a rewrite: “Full texts were of studies included after abstract screening were uploaded….”

We have edited this sentence (line 169).

Line 168 and 169 – you treat data as singular then as plural –please be consistent (and preferably treat as plural)

We have described the data as plural.

Results

Lines 209-2010 – the ‘and’ before ‘three in Western Australia’ is not needed, and ‘in two Victoria’ needs a quick edit

We have edited this sentence (lines 216 – 218).

Line 225 – can you provide more detail on this? E.g., – do you mean they developed theoretical framework or used ones already developed and/or well-established in health behaviour literature? If they used previously developed theoretical frameworks can you provide detail as to which? -assuming some studies used the same or similar frameworks. It may be interesting to know which were most commonly used. Or were these all totally different frameworks?

We have rewritten this sentence to specify theoretical frameworks were take from health literature (lines 232 – 233). Frameworks used were varied, with no particular framework being dominant. Frameworks used have been specified in Table 1.

Table 1 – this is difficult to read given its current presentation with words split across multiple lines.

We have formatted the table to Align Left for easier readability.

Also – in Theoretical Framework – there are both epistemologies and theoretical frameworks in here – I recommend splitting these out so its clear what previously developed theoretical frameworks (e.g., behavioural theories) are being used and, separately, what epistemologies underpin the qual approaches

We are limited in additional space for new columns within this table but have added ‘Epistemology’ to the column title to better describe the content of the column. Further, for 11 of the 27 studies, neither were specified so we think an additional column would take up space without substantially improving the quality of information in the table.  

Table 2 – ‘digital health tackers’ should be ‘digital health trackers’

We have amended this typo error

Section 3.7 – I recommend removing this section text given its brevity but incorporating the statement earlier to highlight the inclusion of geographic information in the previous section and refer the reader to Table 3.

We agree and have moved this text to lines 280 – 282.

Table 3 – formatting issues – column headings – Facilitators and Barriers are not aligned with different columns (i.e., both kind of aligned with the same columns). Is it possible to incorporate this info into table 2? Perhaps by including identifiers next to the statements within the table – e.g., U – urban; RR – regional/rural; R remote – or some similar code so that you can include the key letters next to the statements similar to what you have done with the reference numbers. It might make it easier for the reader to see al of the information, especially given the difficulties with so many columns in table 3 making the text difficult to read easily.

Discussion

Good.

We have formatted the table column headings to be better aligned, and Aligned Left the text, again for easier reading.

We consider Table 3 to provide a digestible synthesis for researchers, policy makers and program implementers who may be seeking evidence specific to urban, regional/rural or remote geographical settings, given the size and great geographical diversity in Australia across areas according to remoteness.